# Reactive Molecular Dynamics Simulations of Polystyrene Pyrolysis

**DOI:** 10.3390/ijms242216403

**Published:** 2023-11-16

**Authors:** Chao Li, Zhaoying Yang, Xinge Wu, Shuai Shao, Xiangying Meng, Gaowu Qin

**Affiliations:** 1College of Sciences, Northeastern University, Shenyang 110819, China; 2100163@stu.neu.edu.cn (C.L.); 2110020@stu.neu.edu.cn (Z.Y.); 2110019@stu.neu.edu.cn (X.W.); 2010023@stu.neu.edu.cn (S.S.); 2Institute of Materials Intelligent Technology, Liaoning Academy of Materials, Shenyang 110004, China; qingw@smm.neu.edu.cn; 3Key Laboratory for Anisotropy and Texture of Materials (MoE), School of Materials Science and Engineering, Northeastern University, Shenyang 110819, China

**Keywords:** polystyrene, activated carbon, pyrolysis, reactive molecular dynamics

## Abstract

Polymers’ controlled pyrolysis is an economical and environmentally friendly solution to prepare activated carbon. However, due to the experimental difficulty in measuring the dependence between microstructure and pyrolysis parameters at high temperatures, the unknown pyrolysis mechanism hinders access to the target products with desirable morphologies and performances. In this study, we investigate the pyrolysis process of polystyrene (PS) under different heating rates and temperatures employing reactive molecular dynamics (ReaxFF-MD) simulations. A clear profile of the generation of pyrolysis products determined by the temperature and heating rate is constructed. It is found that the heating rate affects the type and amount of pyrolysis intermediates and their timing, and that low-rate heating helps yield more diverse pyrolysis intermediates. While the temperature affects the pyrolytic structure of the final equilibrium products, either too low or too high a target temperature is detrimental to generating large areas of the graphitized structure. The reduced time plots (RTPs) with simulation results predict a PS pyrolytic activation energy of 159.74 kJ/mol. The established theoretical evolution process matches experiments well, thus, contributing to preparing target activated carbons by referring to the regulatory mechanism of pyrolytic microstructure.

## 1. Introduction

Activated carbon is an essential industrial product with a wide range of energy, environmental, biology, and medicine applications. The most commonly used raw materials for preparing activated carbon are coconut shells [1], fruit shells, anthracite, bitumen, petroleum coke, etc. [2,3,4,5,6,7,8]. However, even with rigorous modification methods, the preparation of activated carbon from these raw materials still faces the problem of high impurity content, poor performance retention, and uncontrollable pore size. Recently, there has been an increasing number of experimental reports on the fine control of the pore structure of activated carbon using polymers and copolymers, such as polypropylene (PP), polyethylene (PE), polystyrene (PS), polyacrylonitrile (PAN), and styrene–divinylbenzene (S-DVB), as pyrolysis precursors [9,10]. The resulting activated carbon has high purity, regular geometric morphology, and adjustable compositional content [11,12,13,14,15]. Nevertheless, preparing activated carbon from pyrolytic polymer precursors is plagued by unclear mechanisms and difficulties regulating pyrolysis tissues and products.

Polymers are rich in carbon, so converting polymers into high-value-added functional carbon materials is a low-cost process. PS is a typical polymeric material that is widely available, inexpensive, and structurally controllable, and the pyrolysis of PS precursor into functional carbon materials is considered an attractive option for the treatment of PS solids [1,16,17,18]. Moreover, mesoporous carbon has excellent properties as a type of carbon material [19]. In attempts to prepare activated carbon by pyrolysis of PS, it has been found that the heating rate and the target temperature are critical elements in determining the intermediate and final products [1,13,20,21,22]. Since microstructures of carbon materials play a vital role in the properties and characterization, the controlled pyrolysis of polymers has been studied by Tang et al. [14]. There are only a few reports on preparing carbonaceous materials using PS as a pyrolysis precursor [1,13]. Experimentally, although PS thermal decomposition yields relatively simple products compared to other commodity polymers, such as PE [23] and PP [24,25], it is still unsatisfactory in controlling pyrolysis product type, morphology, and pore size [11].

Working towards the controllable preparation of PS pyrolysis, this work adopts the reactive molecular dynamics (ReaxFF-MD) method to study the effects of heating rate and target temperature on the pyrolysis mechanism. The evolution of the microstructure and the generation of pyrolysis products is investigated in detail to provide a clear cognition of the pyrolysis mechanism from the atomic level. It is found that the heating rate would mainly affect the diversity of pyrolysis intermediates, while the target temperature would affect the tissue morphology of the final equilibrium products. Carbon ring analysis bridges theoretical simulations and experimental observations, making ReaxFF-MD simulations practically relevant.

## 2. Results and Discussions

Preparing pyrolyzed carbon materials with well-defined microstructures requires a detailed understanding of the relationship of the pyrolysis mechanism with the heating rate and the target temperature. As is widely recognized, the pyrolysis process consists of two main stages, namely the cleavage of polymers into small organic molecules and the carbonization of degradation products [26,27], and we discuss both aspects below.

### 2.1. Effects of Heating Rate on the Pyrolysis Products

Firstly, the temperature of the PS precursor was raised to 3000 K with different heating rates (5 K/ps, 10 K/ps, 50 K/ps, 100 K/ps, and 200 K/ps), and we examined the effect of different heating rates on pyrolysis intermediates. We found that PS pyrolysis occurs after 2000 K, so Figure 1a shows the trends in the number and types of pyrolysis species during the heating process above 2000 K. According to the changes in pyrolysis intermediates, the heating process can be divided into low-rate (5 K/ps, 10 K/ps) and high-rate heating (50 K/ps, 100 K/ps, 200 K/ps). Overall, the number and type of pyrolysis intermediates at the low-rate heating are more significant than those at the high-rate heating. A peak occurs in the number and type during the low-rate heating, which involves the further cross-linking transformation of intermediate products obtained from the early pyrolysis of PS into larger carbon–hydrogen macromolecules. However, the number and type of pyrolysis intermediates show a negative correlation with the heating rate during the high-rate heating, which indicates that a higher heating rate suppresses the thermal decomposition of the precursor.

Then, 6 ns isothermal simulations at 3000 K were performed for both low- and high-rate heating systems. As Figure 1b shows, in this stage, the total number of pyrolysis intermediates increases while types decrease for every heating rate, which suggests that diverse molecules during the heating process undergo secondary cross-linking reactions that eventually converge to produce the same final pyrolysis equilibrium products. Therefore, the heating rate will not affect the equilibrium carbon products at 3000 K after the isothermal process.

A more detailed pyrolysis products and tissue analysis further proves the above discussion. As shown in Figure 2, hydrocarbon intermediates increase for high-rate heating, while the number and type of hydrocarbon intermediates peak and gradually decrease for low-rate heating. As a result, the heating rate affects the type and quantity of pyrolysis intermediates and their appearance time. In experiments, Onwudili et al. [28] reported the pyrolysis of PS in an intermittent autoclave, where PS was degraded at around 350 °C, and the graphitic carbon increased slightly until 425 °C. The graphitic carbon increased significantly at 450~500 °C [20,22]. In the absence of a catalyst, PS undergoes pyrolysis at temperatures above 500 degrees Celsius, resulting in the production of methane, acetylene, ethylene, styrene, and styrene-like compounds, as well as the formation of graphitic carbon [20,22]. Ahmed et al. [29] studied the differences in PS’ thermal decomposition and gasification characteristics by employing three gradual heating rates of 8~12 °C/min. It was demonstrated that a slower heating rate during the pyrolysis process generates a greater variety of substances. Achilias et al. [30] and Mertinkat et al. [31] discovered, through their experiments on the pyrolysis of polystyrene, that styrene comprises the highest proportion among the pyrolysis products. Thus, the experiments are consistent with our simulations.

### 2.2. Effects of Target Temperature on the Pyrolysis Products

To test the effects of target temperatures on the equilibrium pyrolysis products, we first heated the system to 2500 K, 2750 K, 3000 K, 3250 K, 3500 K, and 3750 K. As described in the previous section, the heating rate hardly affects the equilibrium pyrolysis products, so the 50 K/ps heating rate was implied for every simulated system. Subsequently, a 6 ns isothermal simulation was followed at the respective target temperature, and all simulated systems reached equilibrium, as proved in Appendix A.

The evolution of pyrolysis tissues and products at the isothermal simulation process is shown in Figure 3. According to the changes in microscopic morphology and pyrolysis products, these simulated systems can be classified into a low-temperature system (LTs, 2500 K), proper-temperature system (PTs, 2750 K, 3000 K, 3250 K), high-temperature system (HTs, 3500 K), and ultra-high-temperature system (UHTs, 3750 K).

Comparing the four sets of simulations, we found that the time required for pyrolysis to reach equilibrium is proportional to the target temperature, i.e., the higher the target temperature, the shorter the time for pyrolysis to reach equilibrium. However, the process of pyrolytic evolution and the final equilibrium products are quite different. In the LTs, the equilibrium pyrolysis products are predominantly organized in cross-linked networks with discrete layers of graphite flakes of small molecules. In the PTs, the pyrolysis products show large graphite lamellar tissues and further evolve into randomly folded carbon networks when the pyrolysis equilibrium is reached. In the HTs, large graphite lamellar tissues appear more significantly during the isothermal simulation and evolve into a regular and hierarchical graphitized carbon network at the end of the simulation. However, if the temperature is extremely high, no graphitized tissues are formed, and cracked small molecules are haphazardly dispersed throughout the system at the end of the simulation, as Figure 3 shows. Thus, excessively low and high target temperatures are not conducive to generating large, graphitized tissues.

### 2.3. Tissue Characterization and Carbon Ring Analysis of Pyrolysis Products

In the previous two sections, we discussed the effects of heating rate and target temperature on the pyrolysis intermediates and final products. However, we need a quantitative characterization that can connect with the experiments to discover the microstructure’s evolution during the pyrolysis process, which is crucial for fine-tuning the pyrolysis products in experiments. The previous study showed that the type and quantity of carbon rings in pyrolysis tissues can be fitted by the component peaks in X-ray photoelectron spectroscopy (XPS) [32]. Therefore, in the following section, carbon ring analysis is employed to elaborate on the pyrolysis mechanism.

#### 2.3.1. Carbon Ring Evolution with Different Heating Rates

As described in Section 3.1, the temperature was raised to 3000 K at 5 K/ps, 10 K/ps, 50 K/ps, 100 K/ps, and 200 K/ps heating rates, respectively. The changes in the various types of carbon rings during the heating process at different heating rates are shown in Figure 4. There are three main types of carbocycles generated during pyrolysis, i.e., 5-, 6-, and 7-membered rings, which we denote using 5r, 6r, and 7r, respectively. Since there are only 6r in the PS pyrolysis precursor, the change in its number represents the pyrolysis process, while the presence of 5r and 7r represents the generation of new pyrolysis products.

Figure 4 shows that higher heating rates can inhibit the early cleavage of the 6r and the production of 5r and 7r. Consistent with the previous analysis, this suggests that the low-rate heating (5 K/ps, 10 K/ps) contributes to the sufficient cleavage of the precursor, resulting in the production of type-rich pyrolysis intermediates, as 5r and 7r are significantly observed. In sharp contrast, 6r changes extremely slowly in the high-rate heating plots, with a very small number of pyrolysis intermediates, and no new product formation is observed even at a heating rate of 200 K/ps. The analysis of carbon rings further proves that low-rate heating helps yield more diverse pyrolysis intermediates.

The changes in carbon rings during the next thermostatic simulation at 3000 K are shown in Figure 5. As can be seen, there is a clear inflection point in the 6r plots. At the early stage of the thermostatic simulation, the cracking of the precursor continues and 6r keeps decreasing. During this period, the magnitude of the decrease in the 6r plots is proportional to the rate of heating, i.e., the higher the heating rate, the greater the magnitude of the decrease in the 6r plots, indicating that insufficient cleavage at the heating stage is compensated for in the thermostatic stage.

Subsequently, accompanying the completion of the precursor pyrolysis and the secondary cross-linking reaction, graphite-type tissues are generated in large quantities, as evidenced by the fact that the amount of 6r starts to rise rapidly and becomes the main structure in the final product, outnumbering both 5r and 7r. Since the heating rate hardly affects the equilibrium product, the same equilibrium tissue is obtained at 3000 K for the five heating rates.

#### 2.3.2. Carbon Ring Evolution at Different Target Temperatures

The evolutions of carbon rings in six thermostats with different target temperatures, as defined in Section 3.2, were plotted in Figure 6. We can see that the 6r curves for the other five systems, except for the UHTs (3750 K), show inflection points in the early stage of the isothermal simulations, indicating that further cleavage of the tissues from the heating process is still required at the beginning of the isothermal process. It was found that the higher the target temperature, the faster the further cleavage of the precursor, which is due to the fact that the 6r decline curve becomes steeper with increasing temperature during the initial isothermal process, as shown in the individual subplots. However, due to the excessively high temperature in UHTs, the precursor is completely cleaved, and no inflection point is observed in the system.

After the inflection points, the secondary cross-linking reaction happens, accompanied by a rapid increase in the number of 6r in LTs (2500 K), PTs (2750 K, 3000 K, 3250 K), and HTs (3500 K), and a stable evolution of 5r and 7r. The number of 6r in PTs and HTs is obviously larger than that in LTs. Since graphite has a typical 6-membered ring structure, this change in 6r in the system after the inflection point means that the tissues undergo a graphitizing process, which is sufficient in the PTs and HTs. Regarding the main tissue morphology and the number of carbon rings, the final equilibrium products in the PTs and HTs are similar, consisting of large, graphitized lamellae. However, as will be discussed in the next chapter, the pore size distributions of these tissues are quite different. In the UHTs, the number of 6r carbocycles is almost the same as 5r and 7r, indicating that there is a large amount of discrete fragmented organization in the system and that no large-scale graphite lamellar structure is generated. Therefore, the evolution of the carbon ring under different conditions can be used as a descriptor and measured value by experiments in the controlled pyrolysis process of polystyrene.

#### 2.3.3. Kinetic Analysis of PS Pyrolysis

After establishing the carbon ring characterization of the pyrolysis organization, we can quantitatively analyze the PS pyrolysis process based on the kinetic theory of the reaction. According to Kim’s work [33], the kinetic reaction model of PE pyrolysis can be derived from reduced time plots (RTPs) with the simulation results and then fit to Arrhenius parameters. Our work aims to simulate the preparation of activated carbon by PS pyrolysis, which is microstructured by irregularly cross-linked graphite layers, so the 6r can be used to characterize the conversion of the activated carbon. Therefore, using isothermal simulation data, we statistically analyzed the 6r evolution at various simulated temperatures with time to obtain a kinetic analysis of PS pyrolysis.

The kinetic equation of a solid decomposition reaction can be expressed as follows:(1)$$\frac{d\alpha }{dt}=k\left(T\right)f\left(\alpha \right)=Aexp(-\frac{E_{a}}{RT})f(\alpha )$$
where *α* is the conversion ratio of solid, *f*(*α*) the kinetic model function, and *t*, *T*, *k*(*T*), *A*, *E_a_*, and *R* are the simulation time, temperature, rate constant, pre-exponential factor, activation energy, and ideal gas constant, respectively.

The RTP was constructed by plotting *α* as a function of a reduced time, *t* = *t_α_*, where *t_α_* is the time needed to attain a specific conversion (α = 0.9) at an isothermal simulation temperature *Ti*, and the kinetic equation can be changed to the following integral form:(2)$$\frac{d\alpha }{f(\alpha )}=k\left(T\right)dt$$
(3)$$\int \frac{d\alpha }{f(\alpha )}=\int k\left(T\right)dt$$
(4)$$G\left(\alpha \right)=k\left(T\right)t$$
where *G*(*α*) is the integral form of *f*(*α*). We know that PS pyrolysis at 2500 K, 2750 K, 3000 K, and 3250 K, will produce the same type of activated carbons. Therefore, the model functions listed in Table 1 are used to fit the 6r evolution data at these temperatures (Figure 7a), in which the Mample (first-order) model achieves the highest R-square value.

Subsequently, a linear fit is applied to determine the values of $k\left(T\right)$ on the *G*(*α*)~t relationship at different simulated temperatures, as shown in Table 2. The activation energy is then calculated based on the ln*k*~1/*T* fitted curve, as Figure 7b depicts. The obtained R-square value is 0.94, and the simulated value of *E_a_* is 159.74 kJ/mol, close to the experimental pyrolysis results [33].

#### 2.3.4. Pore Size Analysis

We analyze the pore sizes and their distribution in the respective pyrolysis products to further differentiate the difference in equilibrium tissues at different target temperatures. The pore size distribution in the equilibrium product at each target temperature is shown in Table 3. The detailed evolution of pore distribution can be found in Appendix A.

According to the previous discussion, we know that the LTs and UHTs contain a large number of discretized fragmented molecules, so the conductivity will not be good. We focus on the better-graphitized PTs and HTS systems. Inside the tissues of these systems, 5~11 Å pore size dominates, constituting about 75% of the total. In particular, the 11 Å apertures increase while 5~9 Å apertures decrease with temperature in the PTs. This phenomenon helps to discern tissue nuances in the PTs. It follows that higher temperatures (3250 K) at PTs favor the formation of materials with wider pore size distributions (PSDs).

We should note that the theoretical aperture size correlates with the size of the simulated system. Our simulation box is 25.45 Å^3^; however, the MD simulations of PS pyrolysis are time-consuming and roughly 15 days were used for one isothermal ReaxFF-MD simulation, so we are unlikely to able to test boxes of different sizes. In fact, what we want to know is the correspondence between the pore size obtained from theoretical simulations and the actual pore size. Therefore, after researching the relevant literature [11], we learned that the aperture size obtained by simulation at the scale of our computational model is roughly 25% of the experimental value. In this way, we can learn the pore size of the actual materials.

## 3. Methods and Materials

### 3.1. Modeling of Pyrolysis Precursor

The pyrolysis precursor is modeled by a cubic amorphous polystyrene cell, which contains five 5 PS molecular chains, and the polymerization degree of a single chain is 20, as Figure 8a shows. The monomer of each polystyrene polymer chain is styrene, and a total of 1610 atoms are contained within the assembled regular amorphous unit cell. According to the actual density of polystyrene (1.05 g/cm^3^), the length of the cubic cell is set to 25.45 Å. The first step in performing molecular dynamics simulations is to determine the initial configuration, and a low-energy initial configuration serves as the basis for the simulations. To make the initial structure reasonable, we constructed 10 cell configurations and optimized each configuration with the COMPASS II force field. Then, the configuration with the lowest energy was selected as the precursor model, as depicted in Figure 8b. After modeling amorphous structures, the randomness inherent in the system often leads to significant fluctuations in local energy distributions, even resulting in “vacuum regions” within the structure. Consequently, achieving a stable energy state is not guaranteed, which may hinder subsequent simulations. Therefore, further molecular dynamics optimization and cell relaxation are necessary to find the globally optimal conformation. Thus, an energy minimization of the model was performed before the ReaxFF-MD simulations. In this process, it was geometrically optimized and annealed for simulation. Specifically, 10 cycles of annealing simulations were carried out with the NVT ensemble in the temperature range 200 K~500 K. In this way, the precursor model was further optimized, as Figure 8c shows. We set 10 iterations in this procedure to obtain energetically favorable configurations. The duration of the annealing process typically ranges from 10–100 ps. The total time for the annealing loop is 10 ps. During isothermal simulation, the Andersen thermostat is selected.

### 3.2. ReaxFF-MD Details for Polystyrene Pyrolysis

This work employs NVT integration and high-temperature techniques in ReaxFF-MD pyrolysis simulations on PS with different heating rates and target temperatures. To evaluate the pressure under the NVT ensembles, we plot pressure change with time at various constant temperature simulations in Appendix A. We know that the coordination number of carbon is 3 (graphite structure) at low pressure, and carbon changes to a diamond structure at a pressure of 5 Gpa. In our simulations, the pressure is below the phase transition points, so the pressure will not induce intrinsic changes to the microstructure but may affect the parameters, such as bond length, etc. In addition, the time scale of the experiments is in minutes, while a significant number of reactions in molecular simulations can occur within nanoseconds. Therefore, to accelerate the reaction progress in ReaxFF-MD simulations, higher temperatures are often used to facilitate atomic motion and molecular collisions [34,35], facilitating the observation of reactions [36]. Despite the differences in time and temperature between MD simulations and experiments, numerous studies have shown that increasing the temperature in ReaxFF-MD yields reasonable and consistent results compared to experimental outcomes, and the method has been applied to investigate combustion and pyrolysis in larger systems at high temperatures [36,37,38,39,40].

ReaxFF-MD simulations were carried out with the Large-scale Atomic Molecular Massively Parallel Simulator (LAMMPS) package. In all NVT-ReaxFF-MD simulations, periodic boundary conditions are applied, and the time step is set to 0.2 fs. Temperature control is achieved by the Nose–Hoover thermostat method. We use a stepwise heating method to pre-equilibrate the system at each ramp-up point. Structural energy minimization is performed before the ReaxFF-MD simulations, eliminating any potential strong van der Waals interactions that could lead to local structure distortions and unstable simulations. The minimization (or optimization) procedure is an iterative process in which atomic coordinates are adjusted to minimize the energy of the structure. Snapshots of the simulation results are generated by Visual Molecular Dynamics (VMD, version 1.9.3) [41] and OVITO (version 3.7.10) [42] software at different time points during the process. The pore size distribution is analyzed by the Zeo++ [43] tool. The pyrolysis simulation employs the ReaxFF force field (CHON-2019). This force field was developed to characterize the dissociation and formation of carbon bonds, showing a good agreement with experimental results [44] and, thus, can provide a sound foundation for PS pyrolysis.

## 4. Conclusions

Understanding the heat mechanism is essential to realize controlled pyrolysis of polystyrene. In this paper, we systematically investigated the effect of heating rate and target temperature on the pyrolysis tissues of polystyrene using ReaxFF-MD simulations. The study found that the heating rate would mainly affect the diversity of pyrolysis intermediates, while the target temperature would affect the tissue morphology of the final equilibrium products. From a computational point of view, a temperature increase to 3000~3500 K at a rate of 5~10 K/ps followed by constant temperature pyrolysis yields abundant pyrolysis intermediates as well as an extensive graphitized lamellar tissue. However, constrained by the computational methodology and model size, this by no means implies that the experiments should strictly follow these numbers. Importantly, we have revealed through theoretical simulations the processes that inevitably occur in experiments and are challenging to track in real-time. In this way, by comparing the experimental observations with the theoretical simulations, we can know the pyrolysis stage and decide the direction of the next step, thus, achieving controlled pyrolysis. In the future, we hope to apply this method to prepare amorphous pyrolyzed carbon materials with metallic and nonmetallic loadings, thus, promoting the development of new energy materials.

## Figures and Tables

**Figure 1 ijms-24-16403-f001:**
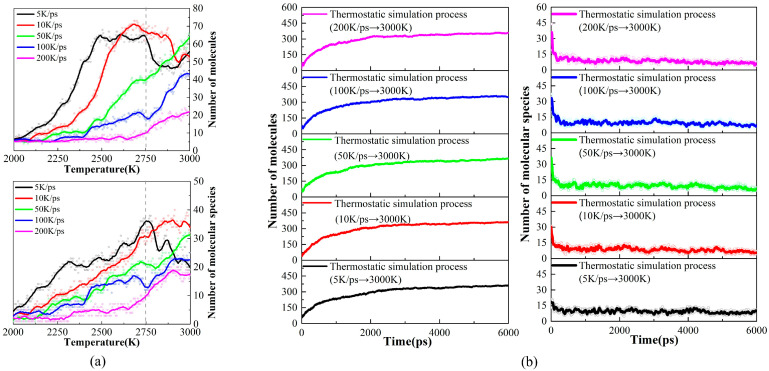
(**a**) Changes in the type and number of pyrolysis products during the heating process; (**b**) changes in the type and number of pyrolysis products during the isothermal process.

**Figure 2 ijms-24-16403-f002:**
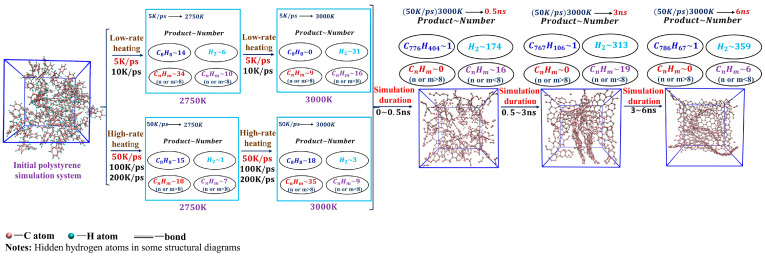
Tissue and species changes during heating and isothermal simulations. To facilitate the comparison of the evolution of the carbon organization during pyrolysis, hydrogen atoms in the polymer are not shown in the figure, since they are released as hydrogen gas in the final pyrolysis products. Detailed hydrogen evolution can be found in the second part of the SI. (Please refer to Appendix A for more details.).

**Figure 3 ijms-24-16403-f003:**
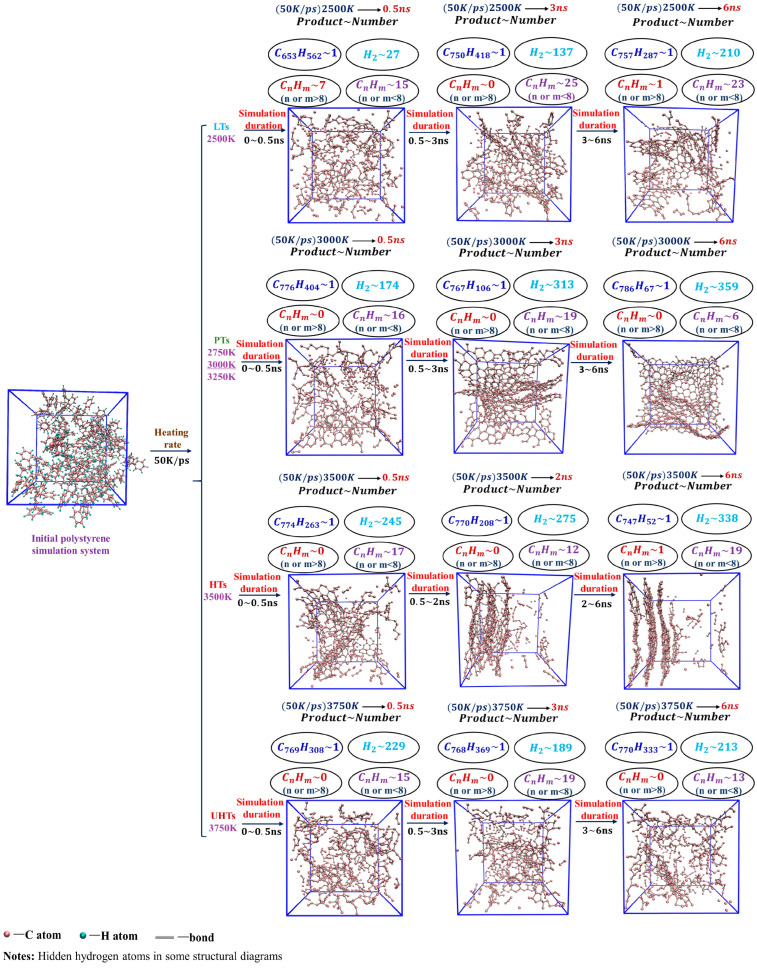
Tissue and species changes during isothermal simulations at different target temperatures.

**Figure 4 ijms-24-16403-f004:**
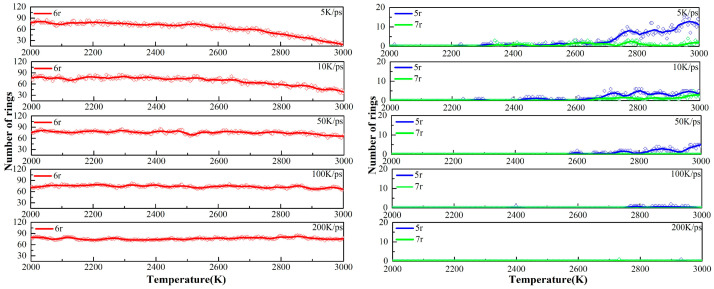
Variation in carbon rings with time during heating to 3000 K at different heating rates.

**Figure 5 ijms-24-16403-f005:**
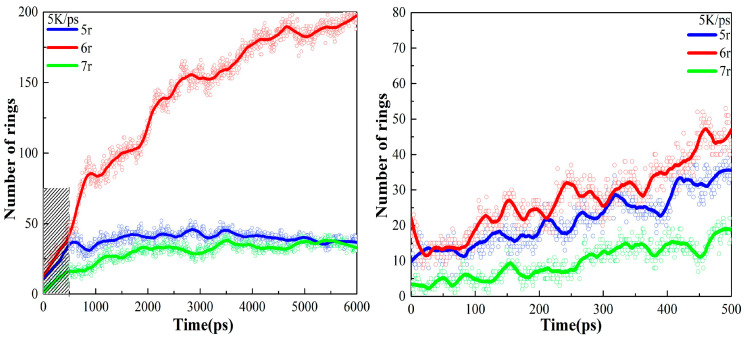
Variation in carbon rings with time during 3000 K thermostatic simulation after heating. The right-hand diagram is an enlargement of the shaded portion of the left-hand diagram.

**Figure 6 ijms-24-16403-f006:**
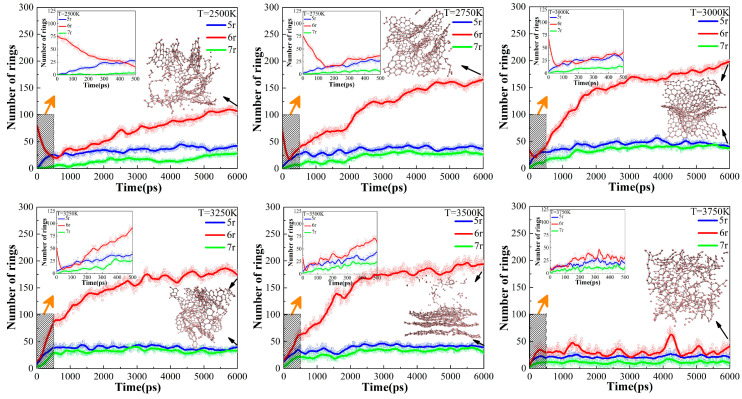
Variation in the carbon rings with time at different constant simulation temperatures. The subplot on the top left enlarges the shaded area.

**Figure 7 ijms-24-16403-f007:**
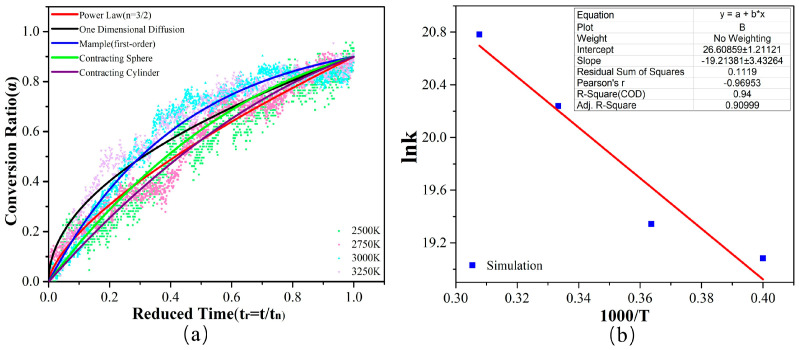
(**a**) Theoretical solid decomposition models and RTP obtained from NVT ReaxFF-MD simulations at 2500 K~3250 K. (**b**) Fitted ln*k* vs. inverse temperature obtained from ReaxFF-MD simulations at 2500 K~3250 K.

**Figure 8 ijms-24-16403-f008:**
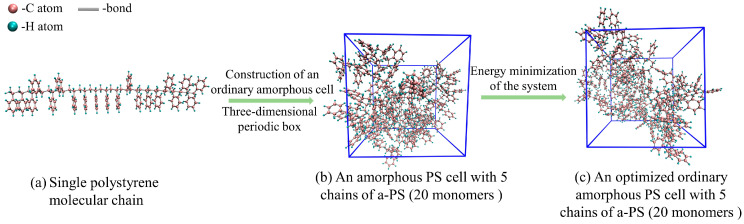
Model of the pyrolysis precursor.

**Table 1 ijms-24-16403-t001:** Differential and integration form of reaction models fitted for the ReaxFF-MD simulation of PS pyrolysis.

Reaction Model	$f\left(\alpha \right)$	G(α)	R-Square
Power law (n = 3/2)	$2/3\alpha^{-1/2}$	$\alpha^{3/2}$	0.8207
One-dimensional diffusion	$1/2\alpha^{-1}$	$\alpha^{2}$	0.9072
Mample (first-order)	1−α	−ln⁡(1−α)	0.9659
Contracting sphere	$3{\left(1-\alpha \right)}^{2/3}$	$1-{\left(1-\alpha \right)}^{1/3}$	0.8809
Contracting cylinder	$2{\left(1-\alpha \right)}^{1/2}$	$1-{\left(1-\alpha \right)}^{1/2}$	0.7959

**Table 2 ijms-24-16403-t002:** The rate constant $k\left(T\right)$ for PS pyrolysis by ReaxFF-MD at different temperatures.

Temperature (K)	$k(s^{-1})$
2750 K	1.94 × 10^8^
3000 K	2.51 × 10^8^
3250 K	6.16 × 10^8^
3500 K	1.06 × 10^9^

**Table 3 ijms-24-16403-t003:** Pore size distribution of final equilibrium products at different temperatures. The unit of the aperture size is Å.

Temperature	Aperture Size	Proportion
2500 K	<5.0	9.0%
	5.0–9.0	44.7%
	9.0–11.5	30.1%
	>11.5	16.2%
2750 K	<5.0	9.1%
	5.0–9.0	55.4%
	9.0–11.5	27.5%
	>11.5	8.0%
3000 K	<5.0	8.3%
	5.0–9.0	41.8%
	9.0–11.5	39.9%
	>11.5	10.0%
3250 K	<5.0	8.4%
	5.0–9.0	40.0%
	9.0–11.5	36.2%
	>11.5	15.4%
3500 K	<5.0	10.0%
	5.0–9.0	42.0%
	9.0–11.5	32.2%
	>11.5	15.8%
3750 K	<5.0	8.9%
	5.0–9.0	44.2%
	9.0–11.5	33.6%
	>11.5	13.3%

## Data Availability

Data are contained within the article and Appendix A [45,46,47].

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
