# Peer review of "Reactive Molecular Dynamics Simulations of Polystyrene Pyrolysis"

_ijms, 2023, doi:10.3390/ijms242216403_

Round 1

Reviewer 1 Report

Comments and Suggestions for Authors

This manuscript investigated polystyrene pyrolysis process using reactive force field molecular dynamics simulation and led to a comprehensive analysis on the heating strategy. It should be published after minor revision.

The major problem is that Figures are way too small to see clearly, especially for Figure 4. Those figures should also be prepared with a higher resolution. 

The second problem is Table 1. The pore size distribution was reported with different bins at different temperatures, making it really hard to compare. I suggest to use a set of unified bins for all temperatures, for example, <5A, 5~7A, 7~9A, 9~11A, and >11A.

Finally, I'm wondering what's the relation between the system size and final pore size distribution. It would be time-consuming to simulate a larger box. But if possible, I do suggest to run simulations with a smaller box to investigate how the pore size distribution changes. 

Comments on the Quality of English Language

Quality of English is good.

Author Response

Response to the reviewer’s comments

Reviewer 1:

Comments and Suggestions for Authors:

1.This manuscript investigated polystyrene pyrolysis process using reactive force field molecular dynamics simulation and led to a comprehensive analysis on the heating strategy. It should be published after minor revision.

Response:

   We sincerely appreciate the reviewer for his/her positive and professional comments on our work, which helps us improve the manuscript's quality.

2.The major problem is that Figures are way too small to see clearly, especially for Figure 4. Those figures should also be prepared with a higher resolution.

Response:

    Thanks for your comment. We have amended it as you suggested.

3.The second problem is Table 1. The pore size distribution was reported with different bins at different temperatures, making it really hard to compare. I suggest to use a set of unified bins for all temperatures, for example, <5Å, 5~7Å, 7~9Å, 9~11Å and >11Å.

Response:

    Thanks for your comment. We have amended it as you suggested.

4.Finally, I'm wondering what's the relation between the system size and final pore size distribution. It would be time-consuming to simulate a larger box. But if possible, I do suggest running simulations with a smaller box to investigate how the pore size distribution changes.

Response:

    Thanks for your comment. The size of our simulation system is 25.452â„« × 25.452â„« × 25.452â„« and the theoretical aperture size correlates with the simulated system's size. However, as you know, the MD simulations of polystyrene pyrolysis are time-consuming and we are unlikely to test boxes of different sizes. In fact, what we want to know is the correspondence between the pore size obtained from theoretical simulations and the actual pore size. Therefore, after researching the relevant literature, we learned that the aperture size obtained by simulation at the scale of our computational model is roughly 25% of the experimental value. In this way, we can learn the pore size of the actual materials.

    We have added the above discussion in the revised text.

Comments on the Quality of English Language:Quality of English is good.

Response:

  Thanks for your professional comments on our work. We are very grateful to you for the constructive suggestion, which has helped us improve the manuscript's quality.

Reviewer 2 Report

Comments and Suggestions for Authors

a.) Authors need to proof two things:

1. the "carbon formation" is not just caused by the system exploded that is due to the very high simulation temperature.

2. A corroboration that what authors saw during simulation are what the experiment have in terms of reactions. Again , this is down to the temperature issue. If one increases the temperature - other reactions may take place. The authors could not proof that this is not the case. One reaxff with RTP simulation using enhanced sampling/metadynamics is required to prove that the reaction at high temperature is the same as at RTP (experiment).

b.) The computational details section is confusing. Moreover, the method in which the authors used to address their scientific question is questionable.

The most important part:

"This work employs NVT integration and high-temperature techniques in ReaxFF-76 MD pyrolysis simulations on PS with different heating rates and target temperatures. 77 Generally speaking, the time scale in experiments is in minutes, while a significant num-78 ber of reactions in molecular simulations can occur within nanoseconds timescale. There-79 fore, to accelerate the reaction progress in ReaxFF-MD simulations, higher temperatures 80 are often used to facilitate atomic motion and molecular collisions [25, 26], facilitating the 81 observation of reactions [27]. Despite the differences in time and temperature between 82 MD simulations and experiments, numerous studies have shown that increasing the tem-83 perature in ReaxFF-MD yields reasonable and consistent results compared to experi-84 mental outcomes, and the method has been applied to investigate combustion, pyrolysis, 85 and explosion in larger systems at high temperatures [27-31]."

c.) Also, please clarify the following items:

1. Figures are stretched out and illegible. I printed it out, cant tell. Especially with Fig 1 and Fig 3. Please change the yellow text on Figure 4. All the texts in bubbles in Figure 4 are illegible. In general, please make all figures bigger.

2. "The pyrolysis precursor is modeled by a cubic amorphous polystyrene cell, which 62 contains five 5 PS molecular chains, and the polymerization degree of a single chain is 20, 63 as Figure 1(a) shows." -- so 100 monomers total in a box? what is the monomer? How many atoms are we talking about? Please add this in the manuscript

3. "To make the initial structure reasonable, we constructed 10 cell configurations" -- what does this means? the authors constructed 10 simulation boxes? why need 10? why not just one and then equilibrate for a while? How did the authors choose which one from the ten they constructed to proceed with? How do you justify and decide of "what is reasonable"?

4. "Specifically, 10 cycles of annealing simulations were carried out with the NVT ensemble in the temperature range 200K~500K." Why need 10 annealing cycle? Please justify the reasoning. How long was each cycle run? What is the total time? And what the temperature that the thermostat was actually set up to?

5. "time step is set to 0.2 fs." -- this is very low. Why did the authors need this slow of a time step?

6. "Firstly, the temperature of the PS precursor was raised to 3000K with different heat-108 ing rates (5K/ps, 10K/ps, 50K/ps, 100K/ps, and 200K/ps), and we examined the effect of 109 different heating rates on pyrolysis intermediates. We found that the PS pyrolysis occurs 110 after 2000K, so Figure 2(a) shows the trends in the number and types of pyrolysis species 111 during the heating process above 2000K." -- this is inconsistent with computational details section that said "annealing from 200-500 K". At which step the 2000K and 3000 K simulation occur?

7. Additionally, how do the authors know that at 2000 and 3000K reactions actually occur, not the simulation box explodes because of too high of kinetic energy that is due to very high temperature? Another proof should be provided, something like a graph of  total force per atom in the box along simulation time.

8. "6ns isothermal simulations at 3000K" this needs to be in the computational details section. Along with all the time for each and total equilibration/annealing/etc. 

9. Figure 2 (b). The letter at the bottom is truncated. Further, Fig2a is very hard to read. More narrative should be provided in the manuscript to guide the reader on what is going regarding Figure 2.  

10. "There-126 fore, the heating rate will not affect the equilibrium carbon products at 3000K after the 127 isothermal process." contradicts "As a result, the heating rate affects the type and quantity of pyrolysis 136 intermediates and their appearance time." Probably just need rewording to make the intention clearer. 

11. "Carbon formation increased slightly until 425°C. 139 However, carbon formation increased significantly at 450~500°C." please define what is "carbon formation'. is not this the content of the simulation box just exploded and give out C atoms?

Comments on the Quality of English Language

english is fine - some sentences probably need rewording, but it is fine

Author Response

Response to the reviewer’s comments

Reviewer 2:

Comments and Suggestions for Authors:

a.) Authors need to prove two things:

1.the "carbon formation" is not just caused by the system exploding that is due to the very high simulation temperature.

Response:

    Thanks for the comment.

    First, we apologize for the misinterpretation caused by the poor use of words. The phrase "carbon formation" appears twice in the original paper, both in the paragraph below Figure 2. The phrase comes from literature, and it describes the experimental products during PE pyrolysis rather than our simulation results. "carbon formation" includes pyrolysis intermediates, such as hydrogen, methane, acetylene, ethylene, etc., and equilibrium products, namely graphitic carbon network, which is not the explosive release of carbon atoms from the system.

    Second, in our simulations, the pyrolysis process involves the cleavage of PE precursors and the secondary cross-linking reaction of the intermediates to form a graphitic carbon network accompanied by hydrogen production. The entire simulation reproduces the chemical reactions in the lab report, proving the validity of the simulation.

    Third, the increased simulation temperature would greatly help observe slow and thermodynamically possible reactions at lower temperatures within pico-second simulations. The simulation technique of “high-temperature short-time simulations are equivalent to low-temperature long-time experiments” has been accepted and widely used [19, 21, 27-31]. In these reports, the pyrolysis systems are often heated to more than 3000 K and calculations are obtained in accordance with the experiment. In our work, to facilitate observation of pyrolysis reactions in acceptable simulation time, the system was heated to 2500~ 3750K, which is within the reasonable simulated pyrolysis temperature range.

    Last, in Figure S1 in the supporting information, we give the temperature and energy change with time during constant temperature simulations. If the simulated system were to explode, there would be an abrupt discontinuous change in the potential energy curve; however, there is no such thing as a legal injury in our curves, but rather the total energy of the system is decreasing as the pyrolysis reaction proceeds and eventually reaches equilibrium.(Please see the attachment for the picture.)

    With these four points, we have justified the temperature settings of the simulated system, and in the revised text we have revised the statements that tend to cause questions and misunderstandings among readers.

2.A corroboration that what authors saw during simulation are what the experiment have in terms of reactions. Again, this is down to the temperature issue. If one increases the temperature - other reactions may take place. The authors could not proof that this is not the case. One reaxff with RTP simulation using enhanced sampling/metadynamics is required to prove that the reaction at high temperature is the same as at RTP (experiment).

Response:

    Thanks for the comment. This question about temperature was partially answered in the last statement. In particular, we would like to thank the reviewers for giving excellent advice on RTP fitting, which will help us to study pyrolysis reaction kinetics.

     Please see the attachment for details.

b.) The computational details section is confusing. Moreover, the method in which the authors used to address their scientific question is questionable.

The most important part:

"This work employs NVT integration and high-temperature techniques in ReaxFF-76 MD pyrolysis simulations on PS with different heating rates and target temperatures. Generally speaking, the time scale in experiments is in minutes, while a significant number of reactions in molecular simulations can occur within nanoseconds timescale. Therefore, to accelerate the reaction progress in ReaxFF-MD simulations, higher temperatures are often used to facilitate atomic motion and molecular collisions [25, 26], facilitating the observation of reactions [27]. Despite the differences in time and temperature between MD simulations and experiments, numerous studies have shown that increasing the temperature in ReaxFF-MD yields reasonable and consistent results compared to experimental outcomes, and the method has been applied to investigate combustion, pyrolysis, and explosion in larger systems at high temperatures."

Response:

    Thanks for the comment. As mentioned earlier, the details of this paper's modeling and computational techniques are frequently used in related pyrolysis simulation studies [19, 21, 27-31]. More importantly, the simulation results are consistent with existing experimental studies. There may have been misrepresentations in the text, which we have corrected in the revised version.

c.) Also, please clarify the following items:

1.Figures are stretched out and illegible. I printed it out, cant tell. Especially with Fig 1 and Fig 3. Please change the yellow text on Figure 4. All the texts in bubbles in Figure 4 are illegible. In general, please make all figures bigger.

Response:

    Thanks for your comment. We have amended it as you suggested.

2."The pyrolysis precursor is modeled by a cubic amorphous polystyrene cell, which contains five 5 PS molecular chains, and the polymerization degree of a single chain is 20, as Figure 1(a) shows." -- so 100 monomers total in a box? what is the monomer? How many atoms are we talking about? Please add this in the manuscript.

Response:

    Thanks for your comment. Inside the ultimate container are five polystyrene polymer chains with a degree of polymerization of 20. The monomer of polystyrene is styrene. The simulated system consists of a total of 1610 atoms.

    We have incorporated this detailed description into the revised paper.

3."To make the initial structure reasonable, we constructed 10 cell configurations" -- what does this means? the authors constructed 10 simulation boxes? why need 10? why not just one and then equilibrate for a while? How did the authors choose which one from the ten they constructed to proceed with? How do you justify and decide of "what is reasonable"?

Response:

    Thanks for your comment. During the process of constructing the ordinary amorphous crystal unit from a molecular chain, the parameter is set to output 10 frames of configurations, rather than constructing 10 simulation boxes. We have performed configuration selection for the ordinary amorphous crystal unit of PS, and selected the frame with the lowest energy for subsequent processes. If the parameter is set to output only one frame of configuration, there is no need for configuration selection. The first step in performing molecular dynamics simulations is to determine the initial configuration, and a low-energy initial configuration serves as the basis for the simulations. Therefore, in order to ensure a reasonable initial structure, we need to select the frame with the lowest energy among the 10 frames of configurations.

    We have made additional clarifications in the revised text.

4."Specifically, 10 cycles of annealing simulations were carried out with the NVT ensemble in the temperature range 200K~500K." Why need 10 annealing cycle? Please justify the reasoning. How long was each cycle run? What is the total time? And what the temperature that the thermostat was actually set up to?

Response:

    Thanks for your comment. After performing the modeling of amorphous structures, the randomness inherent in the system often leads to significant fluctuations in local energy distributions, even resulting in "vacuum regions" within the structure. Consequently, achieving a stable energy state is not guaranteed, which may hinder subsequent simulations. Therefore, further molecular dynamics optimization and cell relaxation are necessary to find the globally optimal conformation. An annealing algorithm, commonly employed, involves gradually increasing the system's temperature followed by a gradual cooling process, aiming to achieve stability. In our case, we have set 10 iterations to obtain a greater number of structural frames. The more final configurations obtained, the better, as this favors the selection of energetically favorable configurations. If feasible, this number can be further increased. The duration of the annealing process typically ranges from 10-100 ps. In our case, the total time for the annealing loop is 10 ps. Forcite utilized a total CPU time of 21 minutes, 55 seconds, and 1314.73s, meaning that each iteration lasted approximately 131.473 seconds. During isothermal simulation, the Andersen thermostat is selected.

    We have made additional clarifications in the revised text.

5."time step is set to 0.2 fs." -- this is very low. Why did the authors need this slow of a time step?

Response:

    Thanks for your question. Our simulation systems contain small-mass atoms, e.g., hydrogen. According to the LAMMPS rules, the time step selection should not only be sufficiently small to capture the fastest dynamics of the system accurately but also large enough to save computational time. Usually, a typical time step is 1fs; however, if the system contains small mass atoms like hydrogen, the step size needs to be set smaller because a larger step would prevent an accurate description of the interactions between molecules. For example, Liu et al. used a time step of 0.25fs in their investigation of the pyrolysis reaction of high-density polyethylene (HDPE) [17].

[17] Zhao, J.; Lai, H.; Lyu, Z.; Jiang, Y.; Xie, K.; Wang, X.; Wu, Q.; Yang, L.; Jin, Z.; Ma, Y.; Hu, Z. Hydrophilic Hierarchical Nitro-gen-Doped Carbon Nanocages for Ultrahigh Supercapacitive Performance. Advanced Materials 2015, 27, 3541-3545.

6."Firstly, the temperature of the PS precursor was raised to 3000K with different heating rates (5K/ps, 10K/ps, 50K/ps, 100K/ps, and 200K/ps), and we examined the effect of different heating rates on pyrolysis intermediates. We found that the PS pyrolysis occurs after 2000K, so Figure 2(a) shows the trends in the number and types of pyrolysis species during the heating process above 2000K." -- this is inconsistent with computational details section that said "annealing from 200-500 K". At which step the 2000K and 3000 K simulation occur?

Response:

    Thanks for your question. The range of 200-500K in the process of constructing the pyrolysis precursor model is dedicated to the annealing step. This step aims to select structures with lower energy for subsequent simulation processes and represents a section of the pyrolysis precursor model construction that is separate from the subsequent heating simulation. After constructing the pyrolysis precursor, we first heat the model at different heating rates, up to 3000K, to investigate the impact of heating rate on the pyrolysis products. We observed that during the heating process of the pyrolysis precursor using different heating rates, there is little production of intermediate pyrolysis products in the simulated system until the temperature reaches 2000K. The pyrolysis primarily occurs in the temperature range between 2000K and 3000K.

    We have made additional clarifications in the revised text.

7.Additionally, how do the authors know that at 2000 and 3000K reactions actually occur, not the simulation box explodes because of too high of kinetic energy that is due to very high temperature? Another proof should be provided, something like a graph of total force per atom in the box along simulation time.

Response:

  Thanks for the comment. We believe the previous answer explains the rationale for the temperature setting. Further, as requested by the reviewer, we give the evolution of kinetic and potential energy over time during the simulation based on the fact that it shows that the system does not explode. (Please see the attachment for pictures.)

8."6ns isothermal simulations at 3000K" this needs to be in the computational details section. Along with all the time for each and total equilibration/annealing

/etc.

Response:

Thanks for the suggestion. We have amended it as you suggested.

9.Figure 2 (b). The letter at the bottom is truncated. Further, Fig2a is very hard to read. More narrative should be provided in the manuscript to guide the reader on what is going regarding Figure 2.

Response:

    Thanks for your comment. The letter at the bottom of Figure 2(b) in the electronic version of our manuscript was not truncated, which is a complete image. The issue you encountered with the truncated letter in the version you saw might have been due to uncontrollable factors.

  We have revised the paper by updating the image and providing an enhanced description for that specific figure.

10."Therefore, the heating rate will not affect the equilibrium carbon products at 3000K after the isothermal process." contradicts "As a result, the heating rate affects the type and quantity of pyrolysis 136 intermediates and their appearance time." Probably just needs rewording to make the intention clearer.

Response:

    Thanks for your suggestion. After constructing the well-heated precursor, the ReaxFF-MD simulation process can be summarized into two main aspects: the initial heating stage and the subsequent constant temperature stage at the desired temperature. Firstly, we conduct the heating simulation with varying rates (5K/ps, 10K/ps, 50K/ps, 100K/ps, 2000K/ps) to raise the temperature of the precursor to a uniform target temperature of 3000K. During this process, we observe the influence of the heating rate on the quantity and variety of intermediate products generated during the thermal decomposition. Subsequently, the aforementioned five sets of structures are subjected to a constant temperature simulation at 3000K for 6ns. Interestingly, we find no discernible differences in the carbon network structures generated by these simulations after the constant temperature phase. Thus, we can conclude that the heating rate does not affect the equilibrium carbon products obtained after the 6ns constant temperature simulation at 3000K. Instead, it primarily influences intermediate thermal decomposition products' types, quantities, and timing during the heating process.

    In our revised manuscript, we provide a more explicit description of this aspect.

11."Carbon formation increased slightly until 425°C. However, carbon formation increased significantly at 450~500°C." please define what is "carbon formation'. is not this the content of the simulation box just exploded and give out C atoms?

Response:

   Thanks for your question. This question is the same as the one we have already answered.

    We have made additional clarifications in the revised text.

Comments on the Quality of English Language:English is fine - some sentences probably need rewording, but it is fine.

Response:

    Thanks for your professional comments on our work. We are very grateful to you for the constructive suggestion, which has helped us improve the manuscript's quality.

Reviewer 3 Report

Comments and Suggestions for Authors

Chao Li and co-workers present a molecular dynamics study using a reactive force field (ReaxFF), in order to rationalise the pyrolysis processes of polystyrene under different heating curves and target temperatures. The main result is that the product distribution (carbon aducts) at equilibrium is a function of temperature. Although this result/trend was expected, the merit lies in the first-principles product characterisation using molecular dynamics techniques. Overall, I think it is a very interesting study, although there are several aspects of the manuscript that need to be improved to make this work publishable.

Major revision:

1) Authors are encouraged to improve the quality of all figures, including font size. All figures must be legible when the manuscript is printed on letter-sized paper. For example, it is virtually impossible to read the numbers in Figures 3, 4 and 7, even at 300% zoom. In addition, some of the colours used, particularly yellow, make them very difficult to read.

2) On line 69, in the section on computational details, the "Forctice force field" is mentioned. I am personally unaware of any force field of this name, and no reference is given, which leads me to believe that this may be a typographical error.

3) It is understood that the generation of initial conditions is performed in the NVT ensemble to introduce a realistic physical state of the system, i.e. to reproduce the PS density. However, in view of the high temperatures treated in the simulations, the use of the same ensemble seems unreasonable. In fact, the experiments performed by Onwudili (reference 41 of the manuscript) show that pressure plays a very important role in the Pyrolysis process, so a comparison with the NPT ensemble seems more appropriate. The authors should provide a justification for the use of the NVT ensemble or, if applicable, show how the results differ from the NPT ensemble for any of the simulation temperatures.

4) The temperatures chosen are intended to accelerate the changes observed in the molecular dynamics simulations, but are too far away from the values at which the pyrolysis reactions are carried out experimentally. What happens if the simulations are performed at the experimental pyrolysis temperature (e.g. 450 K) on a larger time scale? Would there be a difference at 450 K for the NVT and NPT ensembles?

5) The following is mentioned in lines 93 and 94:

    "The minimization (or optimization) procedure is an iterative process in which atomic coordinates and possible cell parameters are adjusted to minimize the energy of the structure."

If the ensemble is NVT, the volume of the simulation box must remain constant. What do the authors mean by the term "cell parameters are ajusted"?

6) It is not very clear from the computational details section whether, once the initial conditions are constructed (after simulated annealing and further optimization), the system is pre-equilibrated at the initial 2000 K temperature of the simulations (or slightly lower). If not, this may have important consequences at the initial stage of interpretation of the results. For example, this would make ambiguous the interpretation about the influence of the heating curve on the formation of intermediates, as it would not be possible to distinguish whether this is a consequence of the different times at which the dynamics is carried out for the different heating curves. Take figure 2a as an example; if there was no pre-equilibration stage, the black (5K/ps) and violet (200 K/ps) curves are not comparable, since the first curve has a dynamics of 200 ps before reaching 3000 K, while the second has a dynamics of only 5 ps.

7) On line 137, the reference to Onwudili is incorrect. In the text it is mentioned as [21] when in the bibliography section it appears as reference [43].

8) In the work of Onwudili et al. cited by the authors, a very comprehensive characterisation of the products of PS pyrolysis is carried out using chromatographic methods. However, the mentions made about this work (lines 137 to 142) focus on very generic aspects, which even appear in the abstract of this reference, instead of giving relevant details that can be compared with the simulation results. On the other hand, the authors comment Westerhout et al. concluded the following (lines 143 and 144):

        Westerhout et al. concluded that even small temperature changes could lead to large changes in reaction rates during PS pyrolysis [38].

However, this mention is somewhat out of context, as it is not a conclusion of Westerhout's paper, but he mentions it as follows  as part of the discussion of a series of simulations (Chem. Eng. Sci., 1996, 51, 2221-2230.):

        Especially the temperature gradients are very important, because pyrolysis reactions have relatively high activation energies and therefore even a small temperature change results in a large change in reaction rate.

In summary, I believe that the comparison with the experiments is only made on a superficial level. What is claimed between lines 144 and 145 of the manuscript requires much more elaboration.

9) In figure 6, the insets for 100K/ps and 200K/ps are not consistent with the main figures. For example, for 100K/ps the left graph shows that the red curve starts at a value of about 25, while the right curve seems at 70. Similar differences can be seen in the other graphs.

My final impression is that the article is more suitable for a journal with a more specialised audience than for IJMC, although I leave this decision to the editor.

Author Response

Response:

Dear Editor,

    We prepared the revision when we received your letter, in which you wrote “Please note that we are waiting for a third report on your paper until Oct. 20. If there is no reply from the Reviewer by that date, we will cancel the invitation and you can revise the paper only based on the already existing reports.”

    Unfortunately, the third reviewer's comment was sent on the 21st, but we did not log in to the submission system to check further after the 20th, so we missed the third reviewer's comment!

    However, based on its presentation, we found that the third reviewer's comments were positive. Moreover, the issues he raised, including the modification of figures and text, the expression of computational simulation methods, and the comparison with experiments, etc., have been corrected in our response to the second reviewer's comments. Therefore, we believe that the revision has responded to the third reviewer's comments.
    So, if you think it's okay, we'll submit it in its current version. Otherwise, please let us know to make further changes.

Sincerely,

Prof. Xiangying Meng

Round 2

Reviewer 2 Report

Comments and Suggestions for Authors

1. Abstract: The word "tissues" is confusing. There is no introduction to the tissue whatsoever.  What tissues? What are we talking about? Needs introduction of the tissue. Introduction of the "tissue" needs to be right there in the abstract, before the first ever "tissue" word was used.

2. Figure 3 and 4 - please change the lime green to something clearer, like the green in the "heating rate"

3. Figure 7 - how is it at 3750 K the carbon forms almost perfect layers while 3250K  shows it is almost of carbon nano tube configuration? Did the tube just collapse? Also, the mono carbon found at the top of the graphite layers, are you sure it is not exploded? I think Fig 7 configuration needs refinement, so it shows one continuous story

4. Figure 8b fonts needs to be bigger

Author Response

Response to the reviewer’s comments

Comments and Suggestions for Authors:

1.Abstract: The word "tissues" is confusing. There is no introduction to the tissue whatsoever. What tissues? What are we talking about? Needs introduction of the tissue. Introduction of the "tissue" needs to be right there in the abstract, before the first ever "tissue" word was used.

Response:

    Thanks for your question. We have corrected the relevant words in the revised manuscript. We replaced the 'tissue' with the 'microstructure'.

2.Figure 3 and 4 - please change the lime green to something clearer, like the green in the "heating rate".

Response:

Thanks for your comment. We have changed the color as you suggested.

3.Figure 7 - how is it at 3750 K the carbon forms almost perfect layers while 3250K shows it is almost of carbon nano tube configuration? Did the tube just collapse? Also, the mono carbon found at the top of the graphite layers, are you sure it is not exploded? I think Fig 7 configuration needs refinement, so it shows one continuous story.

Response:

Thanks for your comment. The almost perfect carbon layer structure in Fig. 7 appeared at the temperature of 3500K, while at the temperature of 3250K, the same spatially folded and curled carbon layer structure was formed in the system as it was at the temperature of 2750K and 3000K. Maybe the viewpoint we chose for the structure caused some confusion for you, so we have re-chosen the viewpoint for the structure picture at 3250K in Fig. 7. Also, we have hidden the hydrogen atoms inside the system to show the carbon structure more clearly. Actually, they are hydrocarbon molecules instead of isolated carbon atoms.

4.Figure 8b fonts need to be bigger.

Response:

Thanks for your comment. We have enlarged it as you suggested.

Reviewer 3 Report

Comments and Suggestions for Authors

The authors did not respond to most of the points raised. While some things were addressed because they were mentioned by the other reviewers (e.g. the quality of the figures), some of the issues I raised were ignored. If I, as a reviewer, make the commitment to read and evaluate your paper (within only one week!), I would expect the same consideration in the response. At least have the courtesy to respond to each point on time.

My consideration of the adequacy of this work has changed, mainly because the inconsistencies in the figures have not been corrected. For example, figure 2a shows that at the end of the heating period, there are less than 20 molecular species for the 200 K/ps curve (purple line). However, in panel 2b, where the isothermal simulations are performed, this value starts above 35. These values should coincide, since in principle, one curve is a continuation of the other. These discrepancies are also noticeable for the 100 K/ps curve (blue line). Similar errors are also present in figure 6. Note that for the curve associated with 200 K/ps for 6r in the right-hand diagram, it is inconsistent with what is observed in the left-hand figure. The red curve in the right panel starts at approximately 70/80, while the left one starts at approximately 25. These inconsistencies are also present in the other curves. 

Author Response

Response to the reviewer’s comments

Comments and Suggestions for Authors:

The authors did not respond to most of the points raised. While some things were addressed because they were mentioned by the other reviewers (e.g. the quality of the figures), some of the issues I raised were ignored. If I, as a reviewer, make the commitment to read and evaluate your paper (within only one week!), I would expect the same consideration in the response. At least have the courtesy to respond to each point on time.

Response:

    We sincerely appreciate the reviewer for his/her positive and professional comments on our work, which helped us improve the manuscript's quality. We apologize for not responding to your previous inquiry in a timely manner due to our failure to check the submission system before the deadline. We sincerely appreciate your understanding and hope that this belated response letter give you a sense of our sincerity.

Comments and Suggestions for Authors:

Chao Li and co-workers present a molecular dynamics study using a reactive force field (ReaxFF), in order to rationalise the pyrolysis processes of polystyrene under different heating curves and target temperatures. The main result is that the product distribution (carbon aducts) at equilibrium is a function of temperature. Although this result/trend was expected, the merit lies in the first-principles product characterisation using molecular dynamics techniques. Overall, I think it is a very interesting study, although there are several aspects of the manuscript that need to be improved to make this work publishable.

Response:

    We sincerely appreciate the reviewer for his/her positive and professional comments on our work, which helped us improve the manuscript's quality.

Major revision:

1) Authors are encouraged to improve the quality of all figures, including font size. All figures must be legible when the manuscript is printed on letter-sized paper. For example, it is virtually impossible to read the numbers in Figures 3, 4 and 7, even at 300% zoom. In addition, some of the colours used, particularly yellow, make them very difficult to read.

Response:

    Thanks for your comment. We have amended it as you suggested.

2) On line 69, in the section on computational details, the "Forctice force field" is mentioned. I am personally unaware of any force field of this name, and no reference is given, which leads me to believe that this may be a typographical error.

Response:

    Thanks for your comment. We have corrected this mistake.

3) It is understood that the generation of initial conditions is performed in the NVT ensemble to introduce a realistic physical state of the system, i.e. to reproduce the PS density. However, in view of the high temperatures treated in the simulations, the use of the same ensemble seems unreasonable. In fact, the experiments performed by Onwudili (reference 41 of the manuscript) show that pressure plays a very important role in the Pyrolysis process, so a comparison with the NPT ensemble seems more appropriate. The authors should provide a justification for the use of the NVT ensemble or, if applicable, show how the results differ from the NPT ensemble for any of the simulation temperatures.

Response:

    Thanks for your comment. To evaluate the pressure under the NVT ensembles, we plot pressure change with time at various constant temperature simulations in Figure 1 (Please see the attachment). Based on the carbon phase diagram, we know that the coordination number of carbon is 3 (graphite structure) at low pressure, and carbon changes to a diamond structure at 5 Gpa pressure. In our simulations, the pressure is below the phase transition points, so the pressure will not induce intrinsic changes to the microstructure but may affect the parameters, such as bond length, etc. In addition, as we list in the references, many molecular dynamics studies on the pyrolysis of organic matter have used NVT systematic simulations, obtaining better results in line with experiments.

   We have added the pressure change graphs to the supporting information and explained the reason for using NVT systematic simulation in the revised text.

4) The temperatures chosen are intended to accelerate the changes observed in the molecular dynamics simulations but are too far away from the values at which the pyrolysis reactions are carried out experimentally. What happens if the simulations are performed at the experimental pyrolysis temperature (e.g. 450 K) on a larger time scale? Would there be a difference at 450 K for the NVT and NPT ensembles?

Response:

    Thanks for your comment. Different reviewers have raised the comment on the simulation temperatures. We would like to prove the reliability of our work from the following points.

    First, as you know, experimental polystyrene pyrolysis is a long time process, but theory does not allow molecular dynamics simulations on the time scale of experiments. To expedite the reaction of the simulated system, we usually adopt the increased simulation temperature to help observe slow and thermodynamically possible reactions at lower temperatures within pico-second simulations. The simulation technique of “high-temperature short-time simulations are equivalent to low-temperature long-time experiments” has been accepted and widely used, as the references [19, 21, 27-31] in the manuscript show. In these reports, the pyrolysis systems are often heated to more than 3000 K and calculations are obtained in accordance with experiments. In our work, to facilitate observation of pyrolysis reactions in acceptable simulation time, the system was heated to 2500~ 3750K, which is within the reasonable simulated pyrolysis temperature range.

    Second, in our simulations, the pyrolysis process involves the cleavage of PS precursors and the secondary cross-linking reaction of the intermediates to form a graphitic carbon network accompanied by hydrogen production. The entire simulation reproduces the chemical reactions in the lab report, proving the validity of the simulation temperature.

    Third, the supporting information gives the temperature and energy change with time during constant temperature simulations (Please see the attachment for the picture). If the temperature is too high to induce system exploding, there would be an abrupt discontinuous change in the potential energy curve; however, there is no such thing as a legal injury in our curves, but rather the total energy of the system is decreasing as the pyrolysis reaction proceeds and eventually reaches equilibrium.

    Finally, as suggested by another reviewer, a reduced-time plot (RTP) has been added to the revised manuscript to compare with experiments to prove that simulations at high temperatures are reasonable.

    With these points, we have justified the temperature settings of the simulated system. In the revised text, we have corrected the statements that tend to cause questions and misunderstandings among readers.

5) The following is mentioned in lines 93 and 94:

"The minimization (or optimization) procedure is an iterative process in which atomic coordinates and possible cell parameters are adjusted to minimize the energy of the structure."

If the ensemble is NVT, the volume of the simulation box must remain constant. What do the authors mean by the term "cell parameters are ajusted"?

Response:

    Thanks for your comment. In the article, we clarified the minimization of structural energy is a step taken before the heating and isothermal ReaxFF-MD simulations. Energy minimization of the model is necessary to rationalize the PS precursor model.  

    The "Minimize" in MD simulation is velocity-independent, meaning it adjusts the interatomic positions to achieve a local minimum energy configuration at a certain temperature.

      In the revised text, we fixed the description of the problem.

6) It is not very clear from the computational details section whether, once the initial conditions are constructed (after simulated annealing and further optimization), the system is pre-equilibrated at the initial 2000 K temperature of the simulations (or slightly lower). If not, this may have important consequences at the initial stage of interpretation of the results. For example, this would make ambiguous the interpretation about the influence of the heating curve on the formation of intermediates, as it would not be possible to distinguish whether this is a consequence of the different times at which the dynamics is carried out for the different heating curves. Take figure 2a as an example; if there was no pre-equilibration stage, the black (5K/ps) and violet (200 K/ps) curves are not comparable, since the first curve has a dynamics of 200 ps before reaching 3000 K, while the second has a dynamics of only 5 ps.

Response:

    Thanks for your comment. Yes, the system has been pre-equilibrated at the initial 2000 K temperature of the simulations. We stress this in the revised text.

7) On line 137, the reference to Onwudili is incorrect. In the text it is mentioned as [21] when in the bibliography section it appears as reference [43].

Response:

    Thanks for your comment. We have amended this mistake in the revised text.

8) In the work of Onwudili et al. cited by the authors, a very comprehensive characterisation of the products of PS pyrolysis is carried out using chromatographic methods. However, the mentions made about this work (lines 137 to 142) focus on very generic aspects, which even appear in the abstract of this reference, instead of giving relevant details that can be compared with the simulation results. On the other hand, the authors comment Westerhout et al. concluded the following (lines 143 and 144):

Westerhout et al. concluded that even small temperature changes could lead to large changes in reaction rates during PS pyrolysis [38].

However, this mention is somewhat out of context, as it is not a conclusion of Westerhout's paper, but he mentions it as follows  as part of the discussion of a series of simulations (Chem. Eng. Sci., 1996, 51, 2221-2230.):

Especially the temperature gradients are very important, because pyrolysis reactions have relatively high activation energies and therefore even a small temperature change results in a large change in reaction rate.

In summary, I believe that the comparison with the experiments is only made on a superficial level. What is claimed between lines 144 and 145 of the manuscript requires much more elaboration.

Response:

    Thanks for your comment. We have made revisions to lines 143-145 in the amended version of the article. The other reviewers also mentioned comparing the experimental results and suggested, "One reaxff with RTP simulation using enhanced sampling/metadynamics is required to prove that the reaction at high temperature is the same as at RTP (experiment)." As a result, we have added section 3.3.3 to the manuscript, providing a more thorough comparison between our simulations and the experimental data.

9) In figure 6, the insets for 100K/ps and 200K/ps are not consistent with the main figures. For example, for 100K/ps the left graph shows that the red curve starts at a value of about 25, while the right curve seems at 70. Similar differences can be seen in the other graphs.

Response:

    Thanks for your comment. We have labeled the bottom of Figure 6 " The right-hand diagram is an enlargement of the shaded portion of the left-hand diagram.".Due to the continuous duration of 6000ps in our isothermal simulation, data was recorded every 0.2ps resulting in a larger number of data points collected. As a result, when plotting and analyzing the final data, the red curve in the full plot on the left (representing the variation in the number of 6-membered rings) is not fully displayed before the "inflection point." Therefore, in order to better observe the change in the number of 6-membered rings at various temperatures before the "inflection point," we have zoomed in on the first 500ps of the isothermal simulation. In other words, the detailed view of the time-dependent trend for the number of 6-membered rings during the first 500ps can be seen in the magnified right image, while the trend after 500ps can be seen in the left image.

    In order to prevent other readers from having the same questions when reading the paper, we have added a description of Figure 6 in the "Supporting Information" to explain the reason for the inconsistency between the starting values of the left and right figures.

My consideration of the adequacy of this work has changed, mainly because the inconsistencies in the figures have not been corrected. For example, figure 2a shows that at the end of the heating period, there are less than 20 molecular species for the 200 K/ps curve (purple line). However, in panel 2b, where the isothermal simulations are performed, this value starts above 35. These values should coincide, since in principle, one curve is a continuation of the other.

Response:

    Thanks for your comment. As you understand, the starting positions of the curves in Figure 2b of the isothermal simulation are an extension of the corresponding colored curves in Figure 2a. Since our thermostatic simulation lasted for 6000 ps and data were recorded every 0.2 ps, we collected a relatively large number of data points finally. Therefore, when drawing, we observed that the changes in the number of molecular species during the initial stage of the constant-temperature simulation were compressed over a short period of time, resulting in a discrepancy in the number of molecular species between the end of the heating process and the start of the constant-temperature simulation. We plotted the local data from the 2500K~3000K heating simulation process against the first 500ps of the constant-temperature simulation. As shown in the figure below (Please see the attachment for the picture), it can be clearly seen that the continuity of the corresponding curve values is obvious.

    In order to prevent other readers from having the same questions when reading the paper, we have added a description of Figure 2 in "Supporting Information" to explain the phenomenon of non-overlapping values.

These discrepancies are also noticeable for the 100 K/ps curve (blue line). Similar errors are also present in figure 6. Note that for the curve associated with 200 K/ps for 6r in the right-hand diagram, it is inconsistent with what is observed in the left-hand figure. The red curve in the right panel starts at approximately 70/80, while the left one starts at approximately 25. These inconsistencies are also present in the other curves.

Response:

    Thanks for your comment. This question is the same as the ninth question in the last review comments. The reviewers have both raised this issue in their feedback from both of the previous reviews, so let us explain here again why.

    We have labeled Figure 6, where the right image is a magnified view of the left image. Due to the continuous duration of 6000ps in our isothermal simulation, data was recorded every 0.2ps resulting in a larger number of data points collected. As a result, when plotting and analyzing the final data, the red curve in the full plot on the left (representing the variation in the number of 6-membered rings) is not fully displayed before the "inflection point." Therefore, in order to better observe the change in the number of 6-membered rings at various temperatures before the "inflection point," we have zoomed in on the first 500ps of the isothermal simulation. In other words, the detailed view of the time-dependent trend for the number of 6-membered rings during the first 500ps can be seen in the magnified right image, while the trend after 500ps can be seen in the left image.

    In order to prevent other readers from having the same questions when reading the paper, we have added a description of Figure 6 in the "Supporting Information" to explain the reason for the inconsistency between the starting values of the left and right figures.

Round 3

Reviewer 3 Report

Comments and Suggestions for Authors

The authors answered most of the points raised. I suggest the editor accept the manuscript for publication in IJMS.